# Epigenetic Regulation and Molecular Mechanisms of Burn Injury-Induced Nociception in the Spinal Cord of Mice

**DOI:** 10.3390/ijms25158510

**Published:** 2024-08-04

**Authors:** Zoltán Mészár, Virág Erdei, Péter Szücs, Angelika Varga

**Affiliations:** 1Department of Anatomy, Histology and Embryology, Faculty of Medicine, University of Debrecen, H-4032 Debrecen, Hungary; meszarz@anat.med.unideb.hu (Z.M.); szucs.peter@med.unideb.hu (P.S.); 2Department of Radiology, Central Hospital of Northern Pest—Military Hospital, H-1134 Budapest, Hungary; erdei.virag6@gmail.com; 3HUN-REN-DE Neuroscience Research Group, H-4032 Debrecen, Hungary

**Keywords:** epigenetics, burn injury, pain, spinal cord, dynorphinergic neuron, histone H3, histone H3 post-translational modification (PTM), p-S10H3, RNA-seq, WES

## Abstract

Epigenetic mechanisms, including histone post-translational modifications (PTMs), play a critical role in regulating pain perception and the pathophysiology of burn injury. However, the epigenetic regulation and molecular mechanisms underlying burn injury-induced pain remain insufficiently explored. Spinal dynorphinergic (Pdyn) neurons contribute to heat hyperalgesia induced by severe scalding-type burn injury through p-S10H3-dependent signaling. Beyond p-S10H3, burn injury may impact various other histone H3 PTMs. Double immunofluorescent staining and histone H3 protein analyses demonstrated significant hypermethylation at H3K4me1 and H3K4me3 sites and hyperphosphorylation at S10H3 within the spinal cord. By analyzing Pdyn neurons in the spinal dorsal horn, we found evidence of chromatin activation with a significant elevation in p-S10H3 immunoreactivity. We used RNA-seq analysis to compare the effects of burn injury and formalin-induced inflammatory pain on spinal cord transcriptomic profiles. We identified 98 DEGs for burn injury and 86 DEGs for formalin-induced inflammatory pain. A limited number of shared differentially expressed genes (DEGs) suggest distinct central pain processing mechanisms between burn injury and formalin models. KEGG pathway analysis supported this divergence, with burn injury activating Wnt signaling. This study enhances our understanding of burn injury mechanisms and uncovers converging and diverging pathways in pain models with different origins.

## 1. Introduction

Pain is an unpleasant experience that can cause tissue damage and inflammation at the periphery and hypersensitivity in the central nervous system (CNS) [1]. At the level of the CNS, all components of the pain system undergo epigenetic, molecular, and functional reorganization following a nociceptive stimulus, which can result in a persistent or chronic pain state [1,2,3]. The neurobiological aspects of nociception are generally well-characterized [1]; however, there is a lack of comprehensive research on the regulation of epigenetics and the molecular mechanisms associated with the pathophysiology of burn injury-induced pain. Epigenetic regulations, such as DNA cytosine methylation, expression of noncoding RNAs, and certain histone modifications (such as acetylation, methylation, and phosphorylation), have been found to play a role in learning and memory formation [4,5], as well as in several CNS disorders [6]. Upon exposure to a noxious stimulus, epigenetic machinery can induce and maintain chronic pain states [2,3,7].

Our understanding of histone modifications induced by burn injury remains limited. The majority of available neuroepigenetic data in pain research are derived from neuropathic pain models, primarily conducted in rats and focusing on the dorsal root ganglion [3,7]. Recent research into the epigenetic regulation of nociceptive circuitry has indicated that pain conditions related to tissue injury, such as burn injury or peripheral formalin administration, lead to increased phosphorylation of histone H3 at serine 10 (p-S10H3) in the ipsilateral dorsal horn of the spinal cord [8,9,10]. Recently, we have shown that CRISPR/cas9-based mutagenesis of histone H3 in the dynorphinergic (Pdyn) neurons of the spinal dorsal horn decreases acute thermosensation [11]. Similar outcomes were observed when the phosphorylation of S10H3 was inhibited by MSK1/2, indicating the importance of the p-S10H3 in pain processing [8]. Therefore, p-S10H3 has been considered to play a significant role in acute pain sensitization. p-S10H3 can attract and interact with other histone H3 modifications, including histone H3 acetylation [7,12]. This interaction between different histone modifications can influence chromatin structure and gene expression. p-S10H3 acts as a bridge, facilitating communication between various epigenetic marks on histone proteins. This indicates the presence of additional epigenetic post-translational (PTM) tags by which nociceptive sensitization is triggered in response to (sub)acute painful stimuli. Therefore, we hypothesized that histone H3 undergoes additional post-translational modifications (PTMs) simultaneously with the phosphorylation of S10H3 in the spinal cord following burn injury. The primary objective of this study was to assess the impact of peripheral burn injury on histone H3 modifications in the spinal cord, utilizing chip-grade histone H3 PTM-specific antibodies.

Our previous findings confirmed that Pdyn neurons, a major component of the endogenous pain-modulating system, play an important role in the development of thermal hyperalgesia after burn injury, at least at the spinal cord level [11]. Consequently, our further objective was to investigate the potential involvement of Pdyn-expressing neurons in the epigenetic alterations induced by burn injury.

A recent study by Mészár et al. [11] shows that the histone H3 mutation at serine 10 does not impact formalin-induced chemo-nociception. This implies that pain models originating from different sources—such as burn injury vs. formalin application—may activate distinct downstream pathways with unique gene expression patterns. To elucidate the pain characteristics induced by burn injury, we performed a genome-wide RNA-seq analysis to investigate the transcriptomic profiles of the lumbar spinal cord in mice following severe scalding-type burn injury. Additionally, the gene expression patterns elicited by burn injury were compared to those induced by formalin-induced inflammatory pain models in adult male mice. This comparative analysis facilitated the identification of differentially expressed genes (DEGs) and the delineation of associated key signaling pathways and functional groups.

Our double immunofluorescence data provide direct morphological evidence highlighting the crucial role of Pdyn neurons in burn injury-evoked pain processing mediated via p-S10H3 in mice. The current study may also help us understand the epigenetic and molecular mechanisms involved in burn injury and provide evidence that the central processing of pain models with different origins converges and diverges along certain pathways.

## 2. Results

### 2.1. Burn Injury Elevates Methylation Levels of Histone H3 at Lysine 4 (H3K4) in the Spinal Cord of Mice as Determined by an Automated Capillary-Based Size Sorting Instrument (WES)

These experiments aimed to investigate the potential involvement of histone H3 post-translational modifications (PTMs) 5 min post-burn injury. To achieve this, histone proteins were extracted from both control and burn injury spinal cord samples. The schematic representation of the experimental workflow is illustrated in Figure 1a. A Simple Western assay using chip-grade antibodies (Table 1) was conducted to examine the presence and relative levels of histone H3 PTMs. This analytical technique provides a sensitive and reliable method for the determination of protein expression levels, enabling a quantitative comparison between the samples. The AUC values generated by the software are directly correlated with the absolute protein amount in the samples. The calculated AUC values were averaged and then compared with those of the untreated control group. Chemiluminescence signals for total histone H3 and various histone H3 PTMs are displayed in a lane view in Figure 2a. Note that in this set of experiments, both ipsi- and contralateral hind paws were exposed to burn injury.

No significant change was observed in the expression of total histone H3 following burn injury at this early time point (5 min post-burn injury; 1.02 ± 0.23-fold that of control AUC; *p* = 0.3; Figure 2b). Upon exposure to thermal pain, histone H3 was found to be significantly hypermethylated at lysine K4, specifically in cases of H3K4me1 and H3K4me3 (Figure 2b; 2.64 ± 0.8-fold and 1.57 ± 0.22-fold that of control AUC, respectively; the respective *p* values were 0.0007 and 0.014). Additionally, there was a marked increase in histone H3K4me3 acetylation at K9 (1.72 ± 0.05-fold that of control AUC; *p* = 0.02).

Neither the overall expression of the dimethylated forms of H3K4 (H3K4me2) nor p-S10H3 was significantly affected by the painful stimulus (Figure 2b; 1.73 ± 0.2-fold and 1.27 ± 0.31-fold that of control AUC, respectively; *p* = 0.2 and 0.17, respectively). While this finding does not necessarily indicate that the burn stimulus has no effect on these histone H3 modifications, it is plausible that the aforementioned PTMs remain hidden due to the high diversity of cell types in the spinal cord. Detailed analysis at the level of distinct cell populations would be crucial to gain a more precise understanding.

### 2.2. A Dual Immunofluorescence Assay Revealed a Notable Increase in the Proportion of Cells Containing Methylated Tags of H3K4 and p-S10H3 after Burn Injury

The schematic representation of the experimental workflow is illustrated in Figure 1b. Histone H3 post-translational modifications (PTMs) were colocalized with the nuclear marker DAPI, demonstrating their expression in the nuclei of the dorsal horn cells of the spinal cord. The results were expressed as the percentage of cells labeled with DAPI (Figure 3a). Note that in this set of experiments, only the ipsilateral hind paw was exposed to burn injury. Quantitative image analyses revealed that 61.4% ± 10.5 (655 out of 1066), 33.4% ± 12.7 (394 out of 1178), 36.4% ± 2.3 (333 out of 914), 38.0% ± 2.9 (488 out of 1284), and 35.5% ± 8.8 (339 out of 954) of DAPI-positive cells were colocalized with H3K4me1, H3K4me2, H3K4me3, and H3K4me3K9ac on the ipsilateral side of noxious heat, respectively (Figure 3a and Appendix A). Contralaterally, the co-expression level between specific histone H3 tags (in the same order as above) and DAPI-stained cells was 49.8% ± 9.3 (515 out of 1034), 20.2% ± 6.5 (246 out of 1214), 27.8% ± 2.4 (286 out of 1027), 23.0% ± 5.5 (303 out of 1312), and 29.2% ± 15.9 (277 out of 880), respectively (Figure 3a and Appendix A).

The proportion of p-S10H3-positive cells within the DAPI-labeled cell population was significantly lower—by approximately 5–10 times—compared to other studied histone H3 markers. Consistent with our previous studies [1,2], 6.09% ± 0.6 (82 out of 1345) of DAPI-positive cells exhibited p-S10H3-immunoreactive (IR) signals in the ipsilateral-sided nuclei, while on the contralateral side, 4.14% ± 1.9 (50 out of 1206) of nucleated cells were p-S10H3-positive (Appendix A). This observation implies that p-S10H3 is a more specific epigenetic marker than the other tags examined.

Based on statistical analysis, the overall expression of burn injury-induced mono-, di-, and trimethylation of H3K4 and p-S10H3 reached statistical significance compared to the contralateral side, regardless of cell type (*p* values are indicated in Appendix A).

### 2.3. Approximately 3%. of Cells in the SDH of Mice Were Identified as Pdyn Neurons

We recently found that dynorphinergic (Pdyn) neurons in the SDH significantly contribute to the development of central sensitization via the p-S10H3-mediated pathway [2,3]. Given that the endogenous opioid system is the primary target of current pain therapy, studying Pdyn neurons seems reasonable based on the above observations. Thus, in the next series of experiments, we studied the colocalization of histone H3 PTMs with Pdyn-expressing neurons in mouse spinal cords using double immunohistochemistry after exposure to noxious heat.

Quantitative analysis of confocal images revealed that nearly 3% of DAPI-labelled cells were Pdyn-immunoreactive neurons in the ipsilateral dorsal horn of the spinal cord from Pdyn::EGFP transgenic mice (number of slices = 21; 230 + 1.04 cells out of 4932 + 12.0 cells; Figure 4). On the contralateral side, 2.5% of the DAPI-labelled cells were identified as Pdyn neurons (n = 21; 194 + 0.8 cells out of 4903 + 18.9; image not shown). Verification of high colocalization between antibodies against neuropeptide precursor preprodynorphin (PPD) and EGFP signals in spinal cord sections from Pdyn::EGFP transgenic mice was previously confirmed [3].

### 2.4. A Double Immunofluorescence Assay Confirmed That Spinal Pdyn Neurons Contribute to Burn Injury-Induced Central Sensitization via the p-S10H3 Pathway

Subsequently, we examined the impact of burn injury on the degree of colocalization between specific histone H3 post-translational modifications (PTMs) and EGFP-labeled Pdyn neurons, quantified as a percentage of the total DAPI-stained cell population. The study aimed to determine the involvement of the dynorphinergic neurons in burn injury-induced epigenetic regulation (Appendix A; Figure 3b,c). Figure 5 displays representative images of transverse spinal cord sections from Pdyn::EGFP mice, demonstrating immunostaining with antibodies targeting EGRP and histone H3 post-translational modifications (PTMs) or total histone H3. Note that only the ipsilateral hind paw was exposed to burn injury in this experiment.

On the ipsilateral side, only 28 out of 1066 dorsal horn cells displayed co-expression of Pdyn and total H3 proteins, accounting for 2.62% ± 0.4, while, on the contralateral side, only 17 out of 1034 cells exhibited this double-labeled neuronal phenotype (1.64% ± 0.15; Figure 3b). On the ipsilateral side, 1.95% ± 0.9 (23 out of 1178) of DAPI-stained cells exhibited colocalization between anti-Pdyn and anti-H3K4me1 antibodies, whereas the contralateral side showed a lower proportion of only 1.07% ± 0.5 (13 out of 1214) of nucleated cells positive for both Pdyn and H3K4me1 (Figure 3b). Double labeling with antibodies against Pdyn and H3K4me2 revealed that 1.85% ± 0.8 (17 out of 914) and 1.65% ± 0.4 (17/1027) of DAPI-labeled cells were positive on the ipsilateral and contralateral sides, respectively. For H3K4me3, the values were 1.79% ± 0.14 (23 out of 1284) and 1.14% ± 0.4 (15/1312), respectively (Figure 3b). Furthermore, 2.2% ± 0.4 of DAPI-positive neurons exhibited colocalization between Pdyn and H3K4me3K9ac antibodies on the affected side, while 1.59% ± 0.9 showed colocalization in the dorsal horn of the spinal cord on the opposite side (21/954 and 14/880, respectively; Figure 3b). On the ipsilateral side, 1.11% ± 0 (15 out of 1345) of nuclei exhibited both p-S10H3 and Pdyn-IR signals, whereas, on the contralateral side, this colocalization occurred in only 0.74% ± 0.3 (9 out of 1206) of nuclei (Figure 3b). This difference reached statistical significance (*p* = 0.012; see Appendix A). Notably, the investigated histone H3 post-translational modification (PTM) showed minimal colocalization with Pdyn-containing neurons relative to the overall population of DAPI-labeled cells following burn injury. Only p-S10H3 exhibited a statistically significant difference among the marks examined between the ipsilateral and contralateral sides (Figure 3b).

Our additional goal was to normalize the co-expression of histone H3 post-translational modifications and EGFP-expressing Pdyn neurons relative to the proportion of Pdyn neurons. This analysis was conducted to elucidate the role of Pdyn neurons located in the spinal dorsal horn in the epigenetic mechanisms induced by burn injury (Figure 3c). Quantitative analysis revealed that 76.9% ± 23.0 (28 out of 34) and 88.6% ± 2.9 (17 out of 19) of Pdyn neurons exhibited co-expression of total histone H3 5 min post-burn injury on the ipsi- and contralateral sides, respectively (Appendix A and Figure 3c). A total of 62.7% ± 7.2 (23 out of 42), 53.84% ± 15.38 (17 out of 30), 74.7% ± 19.9 (23 out of 35), and 82.6% ± 6.2 (21 out of 26) of EGFP-positive Pdyn-neurons were colocalized with H3K4me1, H3K4me2, H3K4me3, and H3K4me3K9ac on the ipsilateral side of noxious heat exposure, respectively (Appendix A and Figure 3c). Contralaterally, the co-expression level between specific histone H3 tags and Pdyn neurons in the same order as above was 37.5% ± 12.5 (13 out of 29), 44.7% ± 1.9 (14 out of 45), 63.7% ± 16.9 (15 out of 25), and 65.7% ± 5.7 (14 out of 22), respectively (Appendix A and Figure 3c). A total of 59% of EGFP-expressing Pdyn neurons had p-S10H3-IR signals on the ipsilateral side, while 31.8% expressed p-S10H3 on the contralateral side (*p* = 0.02; see Appendix A and Figure 3c). This finding is consistent with previously reported results [10].

In summary, we found a high degree of colocalization between EGFP-labeled Pdyn neurons and the examined histone H3 post-translational modifications relative to the proportion of Pdyn neurons following burn injury. However, only the phosphorylated form of the histone H3 at serine 10 (p-S10H3) showed a significant difference. These findings suggest that although there may be an overall increase in histone H3 PTM levels in Pdyn-expressing neurons, only p-S10H3 displayed significant upregulation at least as early as 5 min post-burn stimulus.

### 2.5. There Is Limited Overlap in Differentially Expressed Genes (DEGs) between Burn Injury and Inflammatory Pain Models as Assessed by Transcriptome Analysis

We conducted bulk RNA sequencing to investigate the influence of nociceptive stimuli on the pain-related transcriptional response during the subacute phase. In this set of experiments, we used a formalin-induced inflammatory pain model to compare genes specifically associated with burn damage with genes affected by the inflammation-induced pain model. Figure 1c illustrates the schematic representation of the experimental workflow. Our findings revealed prominent differences in the overall gene expression patterns between animals subjected to painful conditions and their untreated counterparts. Figure 6 and online Appendix A provide further details.

Compared to the non-treated control, with a fold change (FC) threshold of ≥2, we observed upregulation of 50 genes in mouse spinal cord samples following burn injury (BI; Figure 6a), whereas 39 genes exhibited upregulation in response to formalin treatment (FA; Figure 6a). In both pain models, 14 genes exhibited elevated expression at the spinal cord level, corresponding to 28% and 35.8%, respectively. Based on the specified criterion (logFC < −1), we observed the downregulation of 48 genes in the burn injury (BI) samples, along with 47 genes showing downregulation in the formalin-treated spinal cord samples (Figure 6a). Only seven of these genes were affected negatively by both nociceptive stimuli. Next, the pain-associated DEGs in the spinal cord in response to BI and FA mice were used to generate heatmaps by cluster analysis (see below).

#### 2.5.1. Burn Injury-Induced Differential Gene Expression in the Spinal Cord

We identified 98 differentially expressed genes (DEGs) based on the criteria of −1 < logFC < 1. Among them, 50 genes were upregulated, while 48 were downregulated (Figure 6a, Appendix A). Figure 6b,c presents heatmaps illustrating the top 15 genes within each cluster. Detailed information about these DEGs is summarized in Appendix A.

Bulk RNA-seq analysis revealed that the mitochondrial intermembrane space RNase T2 (RNASET2), a pivotal factor in RNA toxicity, demonstrated the highest level of upregulation (fold change value vs. non-treated control: 1.91; Figure 6b left panel; Appendix A). One of the cell adhesion molecules, COMP, which can bind to various cell surface receptors, including integrins, was also overexpressed with a fold change value of 1.63 (Figure 6b left panel). Conversely, MMP28, which degrades components of the extracellular matrix, was decreased by 1.6-fold. (Figure 6c left panel). The receptors for relaxins (RXFP1) and adenosine (ADORA3) exhibited downregulation following burn injury, with respective fold change values of −1.47 and −1.53 compared to controls (Figure 6c left panel). As expected, early response genes that control cell survival and proliferation, such as FOSB and EGR1, were also upregulated in response to noxious heat exposure (Figure 6b left panel and Appendix A).

#### 2.5.2. Formalin-Induced Differential Gene Expression in the Spinal Cord

Following formalin application, we observed upregulation of 39 differentially expressed genes (DEGs) and downregulation of 47 DEGs in the spinal cord samples compared to the untreated samples (Figure 6a, Appendix A). The list of the top 15 upregulated and downregulated genes is displayed in Figure 6b,c. Detailed information about these DEGs is provided in Appendix A.

RNA-seq analysis of spinal cord samples revealed that formalin treatment upregulated mitochondrial genes (MT-ATP8 and MT-CO2) involved in oxidative phosphorylation, with fold change values of 4.14 and 2.43, respectively (Figure 6b middle panel; Appendix A). The NCAPH gene, which plays a crucial role in the positive regulation of chromosome condensation, was significantly overexpressed following formalin injection, with a fold change value of 1.86. Similarly, a transmembrane glycoprotein CD93, involved in various biological processes such as inflammation and immune response [13], was also positively regulated by 1.52-fold vs. control. Given its role in modulating inflammation, CD93 may contribute to pain sensation by influencing inflammatory pathways. In contrast, a decrease in the expression of other inflammation process-associated genes, such as C1QTNF1, was also observed, with a fold change value of −1.68 (Figure 6c; Appendix A). Additionally, a transcriptional regulator, CDCA7, decreased by −1.53-fold in response to formalin.

These findings suggest that formalin might also affect mitochondrial function, chromatin structure, and inflammatory pathways as well.

#### 2.5.3. Common Transcriptional Responses to Burn and Inflammatory Pain in Mouse Spinal Cord

Upon further analysis of the RNA-seq data, it was found that the alpha subunit of hemoglobin, HBA-A2, showed the highest upregulation (Figure 6b right panel). The respective fold change values were 2.92 and 2.94 in wild-type mice in response to burn injury and formalin application. MT-ATP6, involved in the ATP biosynthetic process, was overexpressed, with fold change values of 2.39 and 2.80 in burn injury and formalin treatment, respectively (Figure 6b right panel).

Upon exposure to noxious stimuli, we observed several commonly downregulated differentially expressed genes (DEGs) in the spinal cord, including those involved in chromatin remodeling (CBX2) and metabolic regulation (CAR15; Figure 6c right panel). CAR15, involved in the ATP biosynthetic process, was reduced, with fold change values of −2.39 and −2.80 in burn injury and formalin treatment, respectively. The respective fold change values for CBX2 gene expression were −1.18 and −1.26 in response to burn injury and formalin application. CBX2 can bind to histone H3 trimethylated at Lys-9 (H3K9me3) or Lys-27 (H3K27me3), thereby influencing chromatin structure. Consequently, the downregulation of CBX2 in response to nociceptive stimuli may facilitate the transcription of downstream genes.

Thus, the above data suggested that pain models originating from different sensory modalities converge on common molecular mechanisms underlying subacute pain symptoms. Furthermore, both models display distinct molecular pathways specific to modality-dependent pain processing.

### 2.6. Enrichment Analysis Revealed the Activation of Distinct Molecular Pathways in Burn Injury and Inflammatory Pain Models

Phosphorylation of histone H3 at serine 10 (p-S10H3) causes changes in chromatin structure, making DNA more accessible for gene transcription [9]. Therefore, Gene Ontology (GO) and Kyoto Encyclopedia of Genes and Genomes (KEGG) pathway analyses were performed to assess upregulated pain-related gene clusters in a mouse model of burn injury and inflammatory pain.

Based on the biological relevancy to the experimental models, we constructed a bubble plot representing a subset of biological pathways influenced by burn injury and inflammatory pain evoked by formalin application (Figure 7a). The bubble chart facilitates the identification of both common and divergent pathways in the two pain models, enabling the interpretation of the pathophysiological mechanisms underlying pain induced by various sensory modalities. In the spinal cord, KEGG pathway analyses unraveled that subacute pain processing involves the apelin signaling pathway, arachidonic acid and thiamine metabolism, protein breakdown, etc. Both pain models in mouse spinal cords equally affect signaling pathways regulating focal adhesion, apoptosis, and thermogenesis. Our findings support the significance of PI3K-Akt signaling and ECM–receptor interactions in peripheral inflammation following formalin application. Furthermore, our research indicates a significant upregulation of Wnt signaling pathway genes in response to burn injury. In contrast, exposure to formalin rather triggers the JAK-STAT signaling process (Figure 7a).

While both pain assays activate neuroinflammatory and nociceptive responses, the specific molecular pathways and biological processes differ.

### 2.7. Initial Components of the Wnt Signaling Cascade Are Involved in Burn Injury-Induced Nociception in the Spinal Cord of Mice

Burn stimulus activates nociceptors, leading to the sensation of pain. As we observed in enrichment analysis, the components of the Wnt signaling pathway are likely to play a significant role in this process. Wnt ligands can bind to Frizzled (FZD) receptors on SDH neurons, activating the canonic Wnt/β-catenin pathway [14]. This leads to the regulation of genes involved in pain modulation. Non-canonical Wnt signaling pathways may also contribute to burn injury-induced nociception by influencing intracellular calcium levels and contributing to pain facilitation [14].

DEGs dysregulated by burn injury were mapped onto a KEGG graph using Pathview to demonstrate the impact of burn injury on the Wnt signal pathway (Figure 7b). The KEGG pathway map illustrates the color-coded molecular elements of the typical Wnt-Frizzled-β-catenin signaling pathway that were affected by burn injury. Specifically, Wnt10a, Fzd1, and Sox17a were found to be upregulated, while Ccnd1 and Znrf3 were concurrently downregulated (Figure 7b). Furthermore, Prickle3, which is a transmembrane co-receptor of the FZD protein and a component of the atypical Wnt/PCP signaling pathway, exhibited increased activity in response to burn injury (Figure 7b).

In summary, burn injury predominantly impacts the initial components of both the canonical and non-canonical Wnt signaling pathways, specifically targeting Wnt ligands and their corresponding receptors.

## 3. Discussion

Burn injury leads to specific epigenetic reprogramming, altering functional gene networks associated with pain-associated behavior. Epigenetic tags related to burn injury pain (such as p-S10H3, H3K4me1, H3K4me2, H3K4me3, and H3K4me3K9ac) were elevated in the spinal cord of mice compared to non-treated control. Histone H3 phosphorylation at serine 10 was detected in most Pdyn-containing neurons shortly after burn injury. These findings highlight the role of Pdyn neurons in burn injury-evoked pain processing via p-S10H3. Notably, the differentially expressed genes (DEGs) observed between burn injury and formalin application indicate distinct segregation in pain processing mechanisms between the two groups. Our findings prove that the Wnt signaling pathway is a key player in burn injury, while formalin exposure induces JAK-STAT signaling in mice.

Neurons in the dorsal horn of the spinal cord (SDH) are known to play a key role in the regulation of nociceptive information flow. Epigenomic regulation is essential to these regulatory mechanisms [7]. Epigenetic mechanisms, particularly histone post-translational modifications (PTMs), play a vital role in regulating various biological processes, including neurodegenerative disorders [15,16] and neuropathic pain states [7,17]. In addition, they have been implicated in the pathophysiology of burn injury [8,9,10].

Recent investigations into the epigenetic regulation of nociceptive circuitry revealed that tissue injury-associated pain conditions such as burn injury increased p-S10H3 expression in the SDH neurons [8,9,10]. We have previously shown that CRISPR/cas9-based mutagenesis of histone H3 in the dorsal horn of the spinal cord decreases acute thermosensation [11]. Similar outcomes were observed when the phosphorylation of S10H3 was inhibited by MSK1/2 [8], indicating the importance of the p-S10H3 tag in pain processing. Therefore, at least in its early phase, p-S10H3 has been considered to play a significant role in pain processing. p-S10H3 is a dynamic modification that serves as a recruitment signal for other histone H3 motifs [7,12,18,19]. The concept of a “histone code” was first proposed by Strahl and Allis in 2000 [20]. It suggests that stimulus-dependent modifications occur on one or more N-terminal histone tails, either sequentially or in combination [20]. Hence, investigating the pain-specific “histone code” and subsequent gene expression alterations is of great (pre)clinical significance for understanding the mechanisms of various pain models. Based on the aforementioned concept, it is plausible to hypothesize that pS10H3 is not likely to be the sole modification in the spinal cord of mice following burn injury. However, little is known about additional histone H3 modifications in response to burn injury. To address this issue, histone H3 post-translational modifications induced by burn injury were investigated using high-quality chip-grade antibodies. These antibodies specifically recognize various histone H3 PTMs, including phosphorylation at serine 10, mono-, di-, and tri-methylation at lysine K4, and multiple PTMs on histone H3 (H3K4me3K9ac). While these epigenetic modifications (PTMs) have been associated with the processing of neuropathic pain, as reviewed by Torres-Perez et al. [7], their role in the context of burn injury remains unexplored. However, we cannot rule out the possibility that additional tags on histone H3 or other histone proteins, such as histone H4, may also be implicated in the modulation of tissue damage-associated burn pain.

Our findings indicate that burn injury had an impact on all histone H3 PTMs that were examined. Notably, three out of the five post-translational modifications (H3K4me1, H3K4me3, H3K4me3K9ac) showed significant upregulation in the overall protein levels of the spinal cord. This result aligns with existing reports (see below), and our confocal image analysis also supports this observation. It is important to note that methylation of histone H3 at Lys9 or Lys27 is generally associated with transcriptional repression, while methylation of histone H3 protein at Lys4, Lys36, or Lys79 is typically linked to transcriptional activation [21]. Recent studies have indicated that histone acetylation in the spinal cord plays a significant role in the sensitization of nociceptors, particularly in animal models of neuropathic pain [22,23]. At the onset of a burn injury and during burn-induced tissue repair, there was a change in the acetylation pattern of histone H3 and H4 in the skin in a relevant porcine burn model [24]. The results of the WES study suggest that heightened levels of histone H3 acetylation at Lys9 (K9) may contribute to the development of pain symptoms following tissue damage in a mouse burn injury model. It has been recently demonstrated that the overall level of histone H3 acetylation in rats increases in the spinal cord after burn injury [25]. However, in a different study using a full-thickness burn injury model, rats displayed a significant decrease in histone H3 acetylation at lysine 9 (K9) six hours after lethal burn injury [26]. We have not addressed this discrepancy. Nevertheless, these studies utilized varying antibodies and stimulus intensities (i.e., different burn pain models). The lack of significance in the p-S10H3 level in our WES assay might be due to this epigenetic mark being present in only a small subset of cells. As a result, this modification remains undetectable when analyzing the entire cell population in the spinal cord.

Most of the relevant data about post-translational modifications (PTMs) of histone proteins after tissue injury come from studies on isolated skin tissue of pigs [24], spinal cord tissue [8], and brain samples of rats [26]. Similarly to our study, their conclusions were based on a heterogeneous cellular population consisting of neuronal and non-neuronal cells. Studies published to date lack cell type-specific epigenetic analyses focused on a subset of a certain neuronal population. Our previous findings confirmed that dynorphin-expressing (Pdyn) neurons, which are a major component of the endogenous pain-modulating system, have an important role in the development of thermal hyperalgesia following burn injury, at least at the level of the spinal cord [10]. Dynorphinergic neurons participate in diverse spinal circuits, influencing sensory processing. Specifically, they are involved in gating mechanical sensation [27,28] and chemical itch [29,30], as well as regulating thermal sensitivity [10]. It should be noted that both GABAergic inhibitory and VGLUT2-positive excitatory Pdyn-lineage interneurons exist in the dorsal horn of the spinal cord [10,27,31]. We have previously found that nuclei expressing p-S10H3 predominantly belonged to Pdyn+ neurons following peripheral noxious heat [10]. Within these neurons, the excitatory dynorphinergic neurons, rather than the inhibitory subset, contribute to the processing of acute nociception in burn injury through a p-S10H3-mediated epigenetic process [10]. This finding suggests that the two subpopulations of Pdyn neurons may have distinct functional roles in mediating nociceptive responses to different sensory modalities. Therefore, it would be valuable to investigate the specific contribution of these subgroups of Pdyn neurons to burn injury-induced epigenetic changes and subsequent molecular processes. However, this was beyond the scope of the current study.

Cell-type specificity was investigated using transgenic animals expressing cell-type-specific reporter genes. The Pdyn neuron-specific expression of various histone H3 tags was determined by conventional immunofluorescent staining utilizing transgenic Pdyn::EGFP animals. We found that a majority, potentially up to two-thirds, of the Pdyn neurons showed elevated histone H3 PTMs-IR signals after burn injury. However, significant differences between the ipsilateral and contralateral sides of the spinal cord were observed only for p-S10H3. Our data suggest that Pdyn neurons contribute to the epigenetic modifications associated with burn injury in the mouse spinal cord, which is supported by our previous observations [7,10].

Epigenetic modifications resulting from burn injury can influence gene expression patterns related to the stimulus, affecting sensory perception and the pain experience. The molecular mechanism of various chronic pain conditions was extensively studied in its acute phase (1–2 weeks after pain model establishment) and later in its chronic phase [32,33,34,35,36]. However, the gene expression profiling of the spinal cord following burn injury has not yet been investigated. To the best of our knowledge, this study represents the first comprehensive profiling of gene expression changes and key signaling pathways in the mouse spinal cord following subacute burn injury using bulk RNA-seq. Moreover, we conducted a comparative analysis of gene expression patterns between burn injury and formalin-induced inflammatory pain in mice to gain deeper insights into the molecular pathogenesis of burn injury-induced pain. We aimed to determine whether identical or distinct transcriptional alterations are responsible for the development of pain elicited by various nociceptive stimuli. According to the criteria (−1 > logFC < 1), a total of 98 and 86 differentially expressed genes (DEGs) were identified in the spinal cord after burn injury and formalin injection, respectively. Only a small number of genes overlapped between the experimental groups, indicating that a large proportion of genes were expressed differentially depending on specific sensory modalities.

The list of the top 20 up-and downregulated genes in the L4-L6 spinal cord of mice after burn injury differed from what we detected after formalin-induced inflammatory pain. RNA-seq analysis revealed that the mitochondrial intermembrane space RNase T2 (RNASET2) showed the highest upregulation after burn injury. RNase T2 degrades rRNAs, enhancing nuclear transcription and protein translation through a compensatory mechanism [37]. COMP was found to be overexpressed as a cell adhesion molecule, while MMP28, responsible for extracellular matrix degradation, exhibited reduced expression. The aforementioned alterations have a definite impact on tissue remodeling and the dynamics of the extracellular matrix, potentially contributing to an augmented sensitivity of the nociceptive circuit in tissue damage-associated burn injury. It has been reported that proteases such as MMPs play a crucial role in neuropathic pain by cleaving extracellular matrix proteins (perineuronal net) and/or activating IL-1β, and phosphorylating pERK1/2, which is a marker for central hypersensitization [38,39,40]. The connection between tissue remodeling and noxious stimulus remains incompletely elucidated but is likely to promote a prolonged pain state, probably through mechanisms involving the activation of microglia [41,42]. Furthermore, MMP28 also influences the process of myelination [43]. Burn injury led to downregulation of the relaxin (RXFP1) and adenosine receptors (ADORA3). These transcripts are implicated in neuroprotection by reducing inflammation and tissue damage [44,45]. Another important finding is that early response genes that control cell survival and proliferation, such as Fosb and Egr1, were among the most significantly upregulated transcripts in response to burn injury. Surprisingly, in the inflammatory pain model, they remained unchanged or slightly downregulated. Furthermore, genes such as RASGRP4 and BANK1, which are linked to the positive regulation of the MAPK cascade, ranked among the top 15 upregulated genes affected by burn injury. This result is consistent with our recent findings that upregulation of p-S10H3 expression partially depends on ERK/MAPK downstream signaling in the spinal cord following various nociceptive stimuli [8,9]. Formalin treatment upregulated genes associated with energy production (e.g., MT-ATP8, MT-CO2), chromosome condensation (NCAPH), and inflammation (e.g., CD93) while downregulating genes involved in c-myc-mediated apoptosis (e.g., CDCA7) and the MAPK cascade regulation (e.g., C1QTNF1). The transcript encoded by the C1QTNF1 gene plays a role in the dysregulation of lipid metabolism and inflammatory responses in macrophages during the development of atherosclerosis [46].

Regardless of their origin, both pain models lead to the upregulation of genes involved in mitochondrial function (e.g., HBA-A2, MT-ATP6). Recent transcriptome analysis revealed that hemoglobin genes such as HBA-A2 are significantly upregulated at the cortical level following spinal cord injury [47]. In neurons, hemoglobin may be linked to their need for oxygen and their role in managing oxidative stress. It is believed to help protect against oxidative damage and support mitochondrial function [48]. However, its specific function in neuronal responses to injury is an area of ongoing research. Egr4, a transcription factor, was also found to be overexpressed in response to both painful conditions. Egr4 provides neuroprotection in ischemic stroke by modulating the JNK signaling pathway, as outlined in a recent publication [49]. Genes implicated in chromatin remodeling (CBX2) and the regulation of the apoptotic process (CASP15) exhibited downregulation in the spinal cord in response to noxious stimuli, irrespective of their origin. CBX2 can bind to histone H3 trimethylated at Lys-9 (H3K9me3) or Lys-27 (H3K27me3) [50], thereby influencing the chromatin structure. A recent study demonstrated that CBX2 inhibits MAPK pathway repressive genes via H3K27me3 to activate ERK signaling [50]. Consequently, the downregulation of CBX2 in response to nociceptive stimuli may facilitate the permissive transcription of downstream genes. Interestingly, in chronic pain models, nerve injury resulted in the opposite effect on CBX2 expression [35].

In our study, we conducted a KEGG pathway analysis to examine the upregulated differentially expressed genes (DEGs) in different pain models. We observed that genes associated with the apelin signaling pathway and arachidonic acid metabolism, along with protein degradation processes, exhibited a significantly higher level of upregulation following burn injury as opposed to the formalin model. Both apelin and arachidonic acid (AA) signaling are related to pain modulation and perception in the nervous system with opposite effects. Apelin has an analgesic effect, while the AA derivative prostaglandins exhibit pro-nociceptive activity contributing to pain exacerbation [51,52,53,54]. The KEGG pathway analysis uncovered that PI3K-Akt signaling and ECM–receptor interactions were more prominently associated with the formalin-induced inflammatory pain model than the burn model, which aligns with a previous observation [55]. The two experimental pain models we utilized demonstrated nearly identical effects on focal adhesion, apoptosis, and thermogenesis, as evidenced by the comparable number of affected genes and the significance of their false discovery rate (FDR) values.

The activation of the Wnt signaling pathway is observed in CNS disorders caused by degeneration and neuroinflammation [14]. This pathway is a crucial marker of neuronal inflammation and oxidative stress in CNS diseases and the progression of neuropathic pain [22,56,57,58]. Our research revealed that both typical and atypical Wnt signaling pathways appear to play a role in regulating burn injury in a mouse model, at least in the early phase of noxious input. A previous study by Wu et al. [59] demonstrated the involvement of the Galectin-3 (Gal-3)-dependent Toll-like receptor-4 (TLR-4) pathway in a rat model of full-thickness burn injury, particularly during the later stage of the injury. We observed that burn injury led to the overexpression of Wnt ligands Wnt10a (canonical pathway) and Wnt11 (non-canonical pathway), along with their receptors Fzd1 and Prickle3. Notably, the non-canonical Wnt signal pathway influences the MAPK pathway as well. Prior research has supported the role of the MAPK pathway in burn injury-induced facilitation of the nociceptive circuit (central sensitization) and resulting hyperalgesia [8,9]. In conclusion, burn injury primarily affects the initial elements of both the canonical and non-canonical Wnt signaling pathways, with a specific impact on Wnt ligands and their associated receptors.

Our findings enhance our understanding of central nociceptive processing in the context of burn injury. The current gene expression screening sheds light on the molecular mechanisms behind nociception from diverse sources. We hope that our study will generate exciting new hypotheses and experimental ideas for future pain research. Therefore, it is crucial to continue ongoing research in this field to improve pain management strategies for burn injury-related conditions.

### Limitations, Future Directions, and Clinical Relevance

This study has several limitations. Firstly, the time course of epigenetic post-translational modifications in the spinal cord was not examined; only the 5 min post-injury marks were investigated. Secondly, tissue collection for RNA sequencing was conducted 1 h after the nociceptive stimuli, representing the subacute phase of the pain models. The sampling time was selected based on ethical and animal welfare considerations. Sampling at different time points is likely to result in diverse expression patterns. Thirdly, bulk RNA sequencing lacks the resolution to differentiate between specific nervous system cell types, such as neurons, microglia, and astrocytes. Not only do neurons undergo activity-dependent changes associated with tissue injury, but also glial cells. These changes are also mediated by epigenetic tagging, contributing to the development of pain facilitation, as confirmed by [23,60]. However, the current project did not address the investigation of such changes. Nonetheless, future studies should consider conducting cell type-specific analysis using single-nucleus RNA-seq to obtain comprehensive genetic profiling data of selected subpopulations. Fourth, instead of repeating multiple runs with RNA samples, a single run was conducted using pooled biological samples collected from 12–13 animals in each group. Finally, RNA-seq served as a preliminary/pilot screening method for the transcriptional profiling of experimental models of pain; however, the identified DEGs have not been validated at the protein or functional level.

The current study is fundamental research aimed at identifying potential targets that facilitate the transition from acute pain to chronic pain. Accordingly, the study sheds light on the importance of the Wnt pathway in this process. At this point, it is worth further investigating the possibility of translation, as there are FDA-approved compounds that target the Wnt and beta-catenin pathways [61,62,63]. These compounds are primarily used in cancer therapy (e.g., pimozide). Preclinical testing of these agents in our pain model could provide an immediate therapeutic solution.

## 4. Materials and Methods

### 4.1. Animals and Ethical Considerations

The experiments were carried out on adult male mice. Experiments were approved by the Animal Care and Protection Committee at the University of Debrecen (No.: 23-1/2017/DEMÁB and 15-1/2023/DEMÁB) and were performed according to the European Community Council Directives and the IASP Guidelines. Animals were housed individually in a temperature-controlled colony room and maintained on a 12h/12h light/dark cycle. Food and water were provided ad libitum. We utilized Pdyn-IRES-Cre mice (Pdyncre; JAX #027958, The Jackson Laboratory, Bar Harbor, ME, USA) as a driver strain which expresses Cre recombinase under the direction of the Pdyn (prodynorphin) promoter [64]. These mice were bred with Rosa26-LSL-Cas9 (JAX #026175, The Jackson Laboratory, Bar Harbor, ME, USA) knock-in mice having Cre-dependent expression of CRISPR-associated protein 9 (cas9) endonuclease as well as EGFP reporter protein directed by a CAG promoter [65]. However, the cas9 function was not employed in this study. The resulting offspring (Pdyn::EGFP) displayed EGFP in all Pdyn-expressing neurons. Wild-type C57/Bl6 mice were purchased from Charles River Laboratories (Wilmington, MA, USA).

This study used 44 wild-type C57/Bl6 mice and 5 transgenic Pdyn::EGFP mice.

### 4.2. Establishment of Severe Scalding-Type Burn Injury Model

The used pain model corresponds to the partial thickness burn injury. After administering sodium pentobarbital (Release, WDT, Garbsen, Germany; 50 mg/kg intraperitoneal) to induce deep anesthesia, the left hind leg was immersed in 60 °C water for 2 min, as described previously [9,10,66]. In control cases, animals were left untreated. In the context of RNA sequencing and Simple Western assay (WES), both hind legs were subjected to a burn stimulus concurrently. However, the stimulus was applied solely to the left leg for immunohistochemistry. Depending on the subsequent procedure, animals underwent transcardial perfusion using either 4% paraformaldehyde (PFA) for immunolabeling or sterile saline for the WES study or RNA sequencing (as depicted in Figure 1). These solutions were supplemented with inhibitors from Table 2 to inhibit protein degradation and enhance the stability of histone H3 post-translational modifications (PTMs). After perfusion, spinal cord segments corresponding to L4-L6 at the T12-13 vertebra level were excised and placed on ice after perfusion. In the context of burn injury experiments, a total of 13 wild-type mice were utilized for transcriptome analysis, along with 3 wild-type mice for protein analysis with WES assay. Additionally, 3 Pdyn::EGFP mice were included in the study for colocalization analysis.

### 4.3. Establishment of Formalin-Induced Inflammatory Pain Model

In the formalin-induced inflammatory pain model, 25 μL of a 5% formaldehyde solution was injected intraplanar under the plantar pads of the animal’s hind legs [67]. Both hind paws were exposed to formalin application. This experiment was conducted on 12 wild-type animals under anesthesia using sodium pentobarbital (12 mice; 50 mg/kg i.p.). After one hour of survival, the experiment was terminated without the animal regaining consciousness, and total RNA was extracted from L4–L6 lumbar spinal cords for bulk RNA-seq.

**Table 2 ijms-25-08510-t002:** The reagents of the histone extraction kit (pre-lysis and lysis buffers) and sterile saline were supplemented with additional inhibitors to prevent protein degradation and stabilize histone PTMs caused by burn injury.

Compound	Target	Final cc.	Company #Cat. No	Ref.
Vorinostat (SAHA)	histone deacetylase (HDAC) inhibitor	2 µM	Sigma-Aldrich #SML0061-VAR	[68]
ciclopirox	Pan-histone demethylase inhibitor	100 µM	Sigma-Aldrich #SML2011	[69]
Na-fluoride	protein phosphoseryl- and phosphothreonyl phosphatase inhibitor	0.2% (wt/v)	Sigma-Aldrich #S7920	[8]

### 4.4. Use of Capillary Western Immunoassay (Wes) for Quantification of Histone H3 PTMs Levels

Five minutes after the burn injury, a total of 6 wild-type C57/Bl6 mice (3 control and 3 burn-injured) were subjected to transcardial perfusion with ice-cold sterile saline under deep anesthesia using sodium pentobarbital (see above). Total histone was purified from the bilateral lumbar spinal cord of mice using a histone extraction kit (Abcam #ab113476) in the presence of Halt™ Protease Inhibitor Cocktail (Thermo Fisher Sci., Waltham MA, USA; #78440) and PMSF at a final concentration of 0.3 M (Sigma-Aldrich, Burlington, MA, USA; #P7626). To prevent protein degradation and to stabilize histone H3 PTMs caused by burn injury, solutions (sterile saline, pre-lysis, and lysis buffers of the histone extraction kit) were supplemented with the additional inhibitors listed in Table 2. Protein content was quantified with the Pierce™ BCA Protein Assay Kit (Thermo Fisher Sci. #23225). Protein lysates were pooled separately from control and burn-injured animals. Total histone extraction was diluted to 0.5 mg/mL in sample buffer (manufacturer-supplied reagent), then Fluorescent Master Mix was added in a ratio of 1:4 and incubated at 95 °C for 5 min. The samples were denatured and loaded into WES 13-well plates along with a blocking reagent, primary and secondary antibodies, and a luminol-S and peroxide mixture for separation according to WES instructions. The dilution range of primary antibodies was between 1:10 and 1:100 (see Table 1). Primary antibodies were ordered from Abcam (Cambridge, UK), except for the anti-GFP antibody (Table 1). Cross-reactivity and epitope occlusion were not evaluated due to the antibodies being of chip-grade quality, and thus their reliability was not verified.

Protein separation analysis was conducted using an automated capillary-based size sorting instrument according to the manufacturer’s instructions (WES 3272 system; ProteinSimple, San Jose, CA, USA). A 2–40 kDa Separation Module (ProteinSimple #SM-W010) and Anti-Rabbit Detection Module (ProteinSimple #DM-001) were used. Protein separation and detection were performed by default settings of the instrument (stacking and separation at 375 V for 27 min; blocking reagent for 5 min, primary and secondary antibodies both for 30 min). The chemiluminescence chromatograms generated by the Compass for SW software (4.1.0, Protein Simple) were exported in text format, and the areas under curves (AUCs) representing the absolute quantity of proteins were evaluated in OriginPro 2018 (64-bit SR1, OriginLab Corporation, Northampton, MA, USA) using a peak analyzer. The integration range for evaluation was manually set for each sample (control and burn-injured) to ensure consistency.

### 4.5. Double Immunofluorescent Staining

Two control and three burn-injured transgenic Pdyn::EGFP adult male mice were perfused transcardially with a 4% PFA solution containing some of the inhibitors listed in Table 2 under deep sodium pentobarbital anesthesia. Based on previous protocols [10], L4–L6 segments were dissected, post-fixed for 3 h in PFA, and sectioned with a vibrating blade vibratome (VT 1000S; Leica Biosystems, Nussloch, Germany) at 100 µm thickness. Tris-EDTA buffer (pH 9.0) containing 0.05% Tween 20 was used for antigen retrieval at 95 °C for 30 min. All antibodies were diluted in PBST buffer supplemented with 0.3% Triton-X 100 and 0.3 M NaCl. As a blocking agent, 1% bovine serum albumin (Sigma-Aldrich, #A8531) was added to the primary and secondary antibody mixtures. Details of the primary antibodies applied in this study are presented in Table 1. The primary mixture was incubated for two days at 4 °C, while the secondary mixture was incubated overnight at 4 °C. Three 10 min washes in PBS were performed after incubation with all solutions. Species-specific secondary antibodies were raised in donkey conjugated to Alexa Fluor-555 (Invitrogen, Waltham, MA, USA; #A-31572) and to Alexa Fluor-488 (Invitrogen, #A78948). At the end of the protocol, cell nuclei-specific 4′,6-diamidino-2-phenylindole (DAPI; Sigma-Aldrich #32670) was added to aid in identifying laminar boundaries based on nucleus orientation and density. Sections were mounted in a Hydromount medium (National Diagnostics; Brandon, FL, USA), and confocal images were scanned with Olympus FV3000 confocal systems (Tokyo, Japan).

### 4.6. Total RNA Isolation and RNA-Seq Analysis

Our study aimed to generate comprehensive transcriptome data for three distinct groups of mice: non-treated, burn-injured, and formalin-treated (13, 13, and 12 mice, respectively). To achieve this, we conducted high-throughput mRNA sequencing analysis targeting the spinal cord. The sequencing platform employed for this purpose was the Illumina Sequencing Platform, which allowed us to capture a detailed snapshot of gene expression across these experimental conditions (Illumina Inc., San Diego, CA, USA).

In the treated groups (burn injury and formalin injection), anesthetized mice were exposed to noxious stimuli and subsequently underwent perfusion with ice-cold sterile saline. Treatment was applied to both hind limbs. After 1 h of survival, the bilateral L4-L6 spinal cord segments were removed, washed with ice-cold PBS, and incubated in TRIzol™ Reagent (ThermoFisher Sci, Waltham, MA, USA, #15596026) for 1 h at RT with rotation. After adding chloroform in a one-fifth volume, the crude sample was rotated for an additional 30 min to facilitate lysis. The aqueous phase containing RNA was collected after centrifugation at 12,000× *g* for 30 min at 4 °C. After being treated with ice-cold isopropanol, the sample was centrifuged for 30 min at 20,000× *g* at 4 °C. The total RNA precipitate was dissolved in 80% ice-cold ethanol in RNase-free water and then centrifuged at 20,000× *g* for 10 min at 4 °C. The air-dried RNA pellet was resuspended in 30 µL of RNase-free water by pipetting. The concentration of RNA was determined using NanoDrop, and RNA samples from individual animals were pooled with equal RNA amounts into three groups according to the treatments. Finally, according to the manufacturer’s protocol, the pooled RNA quality was checked on an Agilent BioAnalyzer using the Eukaryotic Total RNA Nano Kit (Agilent Technologies, Santa Clara, CA, USA). All the samples had an RNA integrity number (RIN) value >8; thus, they were accepted for the library preparation. According to the manufacturer’s protocol, RNA-seq libraries were prepared from total RNA (200 ng) using the NEBNext^®^ Ultra II RNA Sample Preparation Kit for Illumina (New England BioLabs, Ipswich, MA, USA). Sequencing runs were executed on the Illumina NextSeq500 instrument (Illumina) using single-end 75-cycle sequencing.

Raw sequence data (.bcl files) were converted into fastq files and demultiplexed using Illumina’s bcl2fastq software. Data were aligned and normalized using a STAR aligner (ver 2.7.11, [70]) and mapped against the mouse genome (Ensembl, GRCm39). Differentially expressed genes were identified by edgeR (ver 3.18) according to [71]. We used the appropriate R packages (VennDiagram and ggplot2, ver 4.3.3 for both, https://www.R-project.org) and SRPLOT [72] to visualize gene expression data. Gene pathway analysis was performed using Cytoscape (ver 3.10.1, [73]) utilizing its built-in String (ver 2.0.2) and Enrichmentmap (ver 3.3.6) apps. Aligned sequencing data have been deposited into the NCBI SRA database under accession number PRJNA1137110.

### 4.7. Data Analysis

All data are expressed as the mean ± standard error of the mean (SEM). Statistical significance was determined using Student’s *t*-test to compare the means between different groups. In the case of Gene Ontology KEGG pathway analysis, the false discovery rate (FDR) was applied to express how strongly correlated the found genes in our dataset were with all genes described in a given pathway using the built-in algorithm of the Enrichmentmap app of Cytoscape.

## 5. Conclusions

Peripheral burn injury leads to specific epigenetic reprogramming, modifying functional gene networks and contributing to pain-related behavior. We found that burn injury pain-associated epigenetic tags (i.e., p-S10H3, H3K4me1, H3K4me2, H3K4me3, and H3K4me3K9ac) were elevated in the spinal cord of mice compared to the non-treated sample, using chip-grade antibodies. Significant hypermethylation of histone H3 at lysine 4 was detected in a mixed population of spinal cord samples 5 min after burn injury. This observation is consistent with our results from confocal image analysis. In the context of burn injury, we observed that all examined histone H3 post-translational modifications (PTMs) exhibited elevated levels in DAPI-labeled cells on the affected side compared to the contralateral SDH. However, only p-S10H3-immunoreactivity exhibited a significant increase in Pdyn neurons. These findings provide direct morphological evidence highlighting the crucial role of Pdyn neurons in burn injury-evoked pain processing mediated via p-S10H3 in mice.

A subset of differentially expressed genes (DEGs) exclusively associated with burn injury was identified by comparing the transcriptomic profile of burn injury with inflammatory pain-related DEGs. Only a limited number of overlapping up- and downregulated DEGs were detected between burn injury and formalin application, suggesting notable differences in the central processing of pain models with different origins, at least at the subacute stage. The analysis of KEGG pathways provides further support for this concept. Burn injury increases the activity of genes in the Wnt signaling pathway, while exposure to formalin triggers the JAK-STAT signaling process. Our findings revealed several genes and pathways associated with burn injury that may be targets for further pain research.

## Figures and Tables

**Figure 1 ijms-25-08510-f001:**
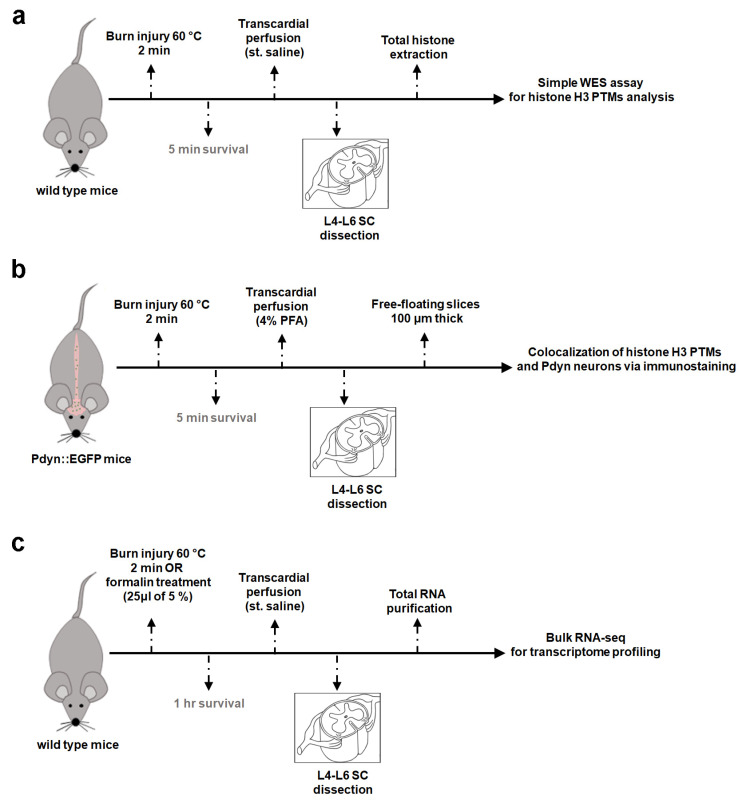
Simplified experimental workflow. (**a**) Capillary Western immunoassay (WES) for quantification of histone H3 PTM protein levels. (**b**) Colocalization analysis of histone H3 PTMs and Pdyn neurons via immunofluorescent labeling. (**c**) Bulk RNA-seq for transcriptome profiling. All treatments were performed on anesthetized mice, ensuring that the animals did not regain consciousness. In the context of burn injury, six wild-type C57/Bl6 mice were used for protein assay (WES), five Pdyn::EGFP mice for colocalization analysis, and thirteen wild-type mice for RNA-seq. In addition, 12 wild-type mice were exposed to formalin for further RNA-seq. Please note that for the immunostaining study, only the left hind paws were subjected to burn injury (**b**), while in other experiments (**a**,**c**), treatments were bilateral. WES, capillary Western immunoassay; SC, spinal cord.

**Figure 2 ijms-25-08510-f002:**
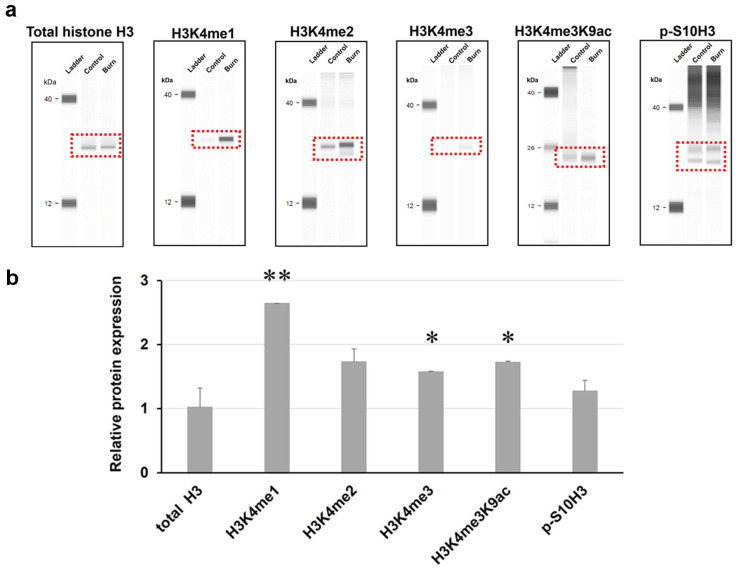
Burn injury results in hypermethylation of histone H3 at lysine 4 (H3K4) in the spinal dorsal horn (SDH) of mice as determined by an automated capillary-based size sorting instrument (WES). (**a**) Following histone extraction from control and burn injury spinal cord samples, several histone H3 PTMs were measured by Simple Western assay using chip-grade antibodies (n = 3 mice per group). Representative chemiluminescence bands generated by the Compass for Simple Western (SW software 4.1.0, Protein Simple) are displayed here. Bands enclosed by red squares indicate specific post-translational modifications for histone H3 (see Table 1). (**b**) The protein levels were compared to those of the non-treated group. Quantification of panel (**a**) data, mean ± SEM (n = 6 replicates, * *p* < 0.05, ** *p* < 0.01 vs. non-treated control group; Student’s *t*-test). Note that both hind paws were exposed to burn injury in this experiment.

**Figure 3 ijms-25-08510-f003:**
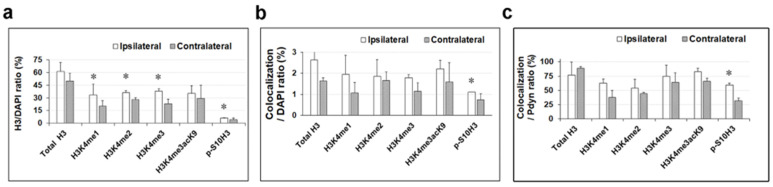
A dual immunofluorescence assay revealed a notable increase in the proportion of cells containing methylated tags of H3K4 and p-S10H3 after burn injury. Quantitative analyses of immunoreactions after burn injury, mean ± SEM, * *p* < 0.05 vs. non-treated control group (Student’s *t*-test). Panel (**a**) illustrates the impact of burn injury on the expression levels of total histone H3 and specific histone H3 post-translational modifications (PTMs), represented as a percentage relative to DAPI-labeled cells. Panel (**b**) displays the effect of burn injury on the colocalization level between histone H3 PTMs and EGFP-expressing Pdyn neurons, normalized in relation to DAPI-labeled cells. Panel (**c**) illustrates the impact of burn injury on the level of colocalization between histone H3 PTMs and EGFP-expressing Pdyn neurons, presented as the % of Pdyn neurons.

**Figure 4 ijms-25-08510-f004:**
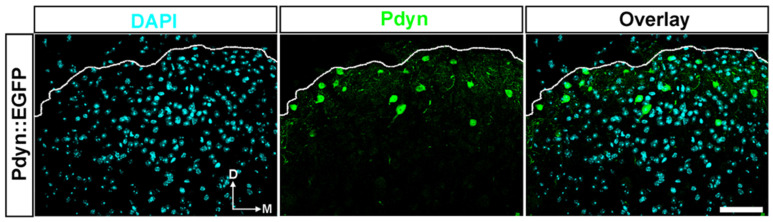
A notable portion of cells in the dorsal horn of the spinal cord in transgenic mice were identified as Pdyn neurons. Representative images showing immunostaining with antibodies against EGRP (green) and DAPI (magenta) in the dorsal horn of the spinal cord from a Pdyn::EGFP mouse. Quantitative analysis revealed that nearly 3% of nucleated cells were Pdyn-immunoreactive neurons. The dotted line indicates the border between gray and white matter. D, dorsal; M, medial. Scale bars, 50 μm.

**Figure 5 ijms-25-08510-f005:**
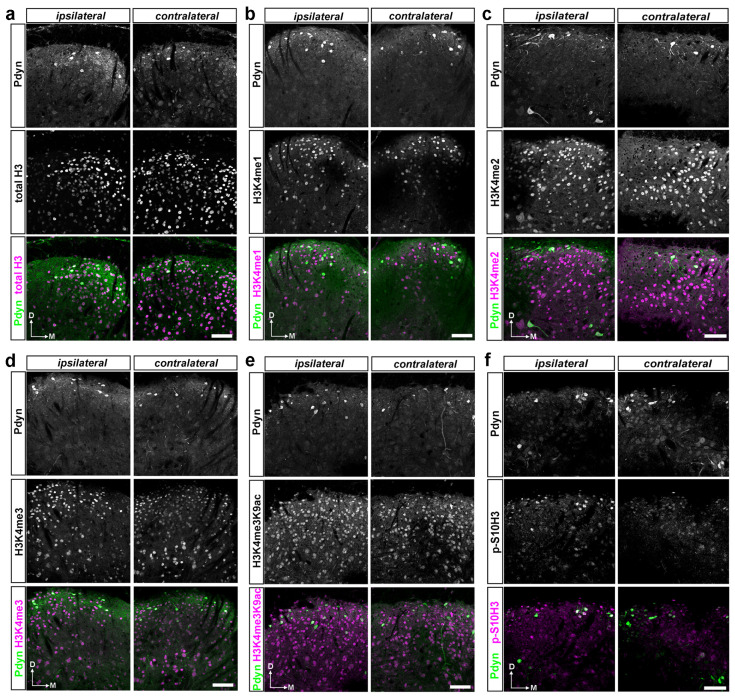
A dual immunofluorescence assay on free-floating spinal cord sections confirmed that spinal Pdyn neurons contribute to burn injury-induced epigenetic reprogramming, particularly via the p-S10H3 pathway. (**a**–**f**) Representative images display immunostaining with antibodies against EGRP and histone H3 PTMs or total histone H3 in transverse spinal cord sections from Pdyn::EGFP mice, where all Pdyn neurons express EGFP. Individual channels are presented in greyscale, and pseudocolor is used in the composite images. Note that in this set of experiments, only the left hind paw (ipsilateral) was exposed to burn injury. Scale bars, 50 μm.

**Figure 6 ijms-25-08510-f006:**
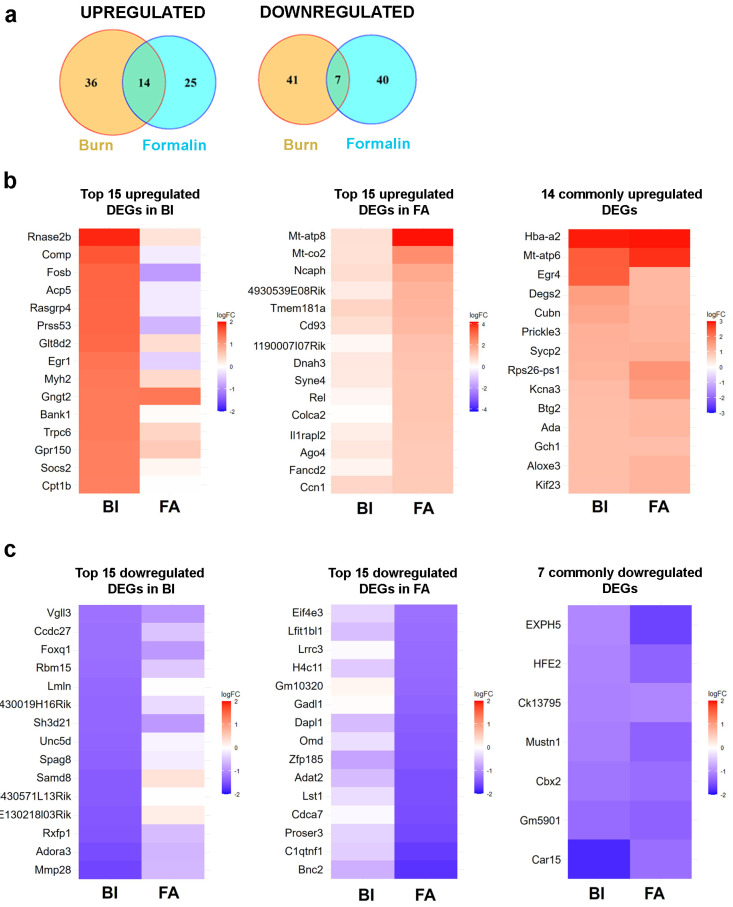
RNA sequencing analysis revealed that there is limited overlap in differentially expressed genes (DEGs) in the spinal cord between the burn injury and inflammatory pain model. (**a**) Venn diagram analysis illustrating the overlap of differentially expressed genes (DEGs) (−1 < logFC < 1, compared to the untreated sample) in the spinal cord between the burn injury model and an inflammatory pain model induced by the application of formalin. The intersections of circles represent DEGs commonly dysregulated in response to noxious exposures. (**b**) Heatmap of RNA-seq data showing the top 15 upregulated differential expressed genes (DEGs) in the spinal cord following burn injury (BI on the left panel) and formalin application (FA in the middle panel). The panel on the right lists the 14 DEGs that are upregulated in both pain models. In each panel, color-coded expression patterns of the identical genes are shown side by side to facilitate a comparative analysis of the two pain models. The blue color indicates downregulated genes, while warm colors represent upregulated genes. The molecular functions and biological processes involved in the top 15 upregulated DEGs are detailed in Appendix A. (**c**) Heatmap of RNA-seq data showing the top 15 downregulated DEGs in the spinal cord following burn injury (BI on the left panel) and formalin application (FA in the middle panel). The right panel lists the 7 DEGs that are downregulated in both pain models. In each panel, color-coded expression patterns of the identical genes are shown side by side to facilitate a comparative analysis of the two pain models. The blue color indicates downregulated genes, while warm colors represent upregulated genes. The molecular functions and biological processes involved in the top 15 downregulated (DEGs) are detailed in Appendix A. BI, burn injury; FA, formalin application.

**Figure 7 ijms-25-08510-f007:**
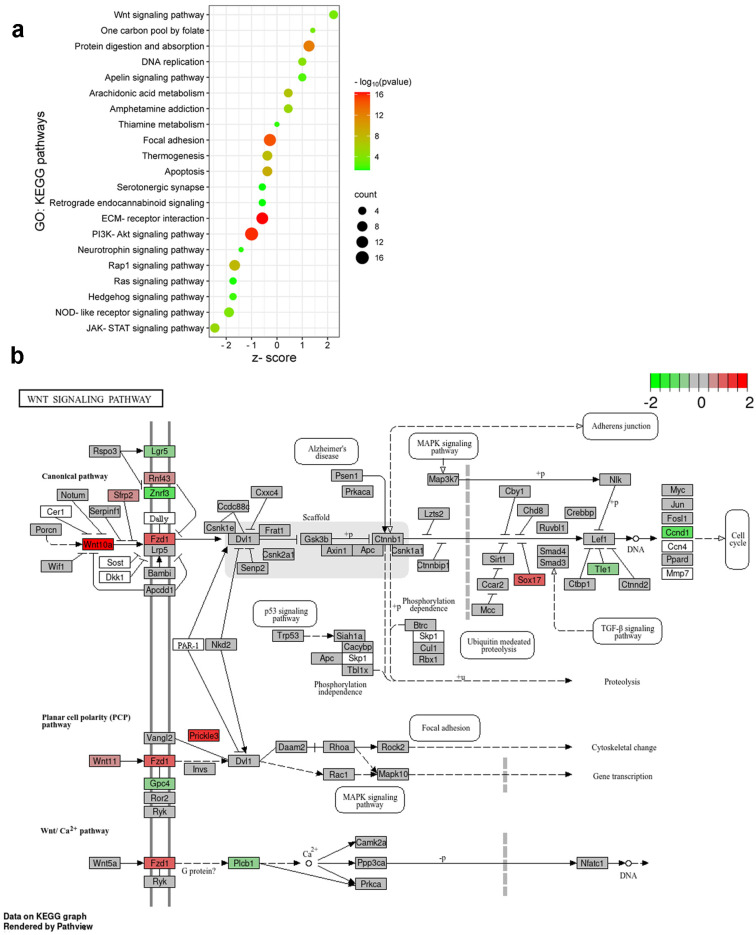
Distinct molecular pathways activated by burn injury and formalin application. (**a**) Bubble plot represents the Gene Ontology (GO): Kyoto Encyclopedia of Genes and Genomes (KEGG) pathways activated in spinal cord samples following FA and BI. The pathways were selected based on their relevance to nervous tissue and the upregulated genes identified in the burn injury or formalin experiments (refer to the heatmap images in Figure 6). These upregulated genes and their second-order interacting genes (filtered by a log fold change criterion of −0.5 < log2FC > 0.5) form the core of our molecular network. This second-order molecular network and the KEGG pathway search were conducted using the Cytoscape String app (version 3.10.1, https://string-db.org/). The size of each bubble corresponds to the number of genes associated with a given pathway. The z-score value indicates the ratio of these genes upregulated in the inflammatory pain model (negative values) or in burn injury (positive values). The color gradient represents the false discovery rate (FDR) on a log10 scale, providing a measure of statistical significance. (**b**) The initial components of the Wnt signaling cascade are involved in burn injury-induced nociception. KEGG Pathview analysis of the molecular signature of the Wnt signaling pathway in burn injury vs. control. The KEGG pathway analysis shows that burn injury results in the upregulation or downregulation of multiple components of the Wnt signaling pathway. Each box is one gene. The color (in the log2 scale) represents the log2-based fold change. Gene expression levels are indicated as higher (red), unchanged (gray), or lower (green) in burn-injured mice compared to non-treated control.

**Table 1 ijms-25-08510-t001:** List of the primary antibodies used (catalog number, dilution factors for various applications, and expected/observed band sizes in the WES assay).

Primary Antibody	Abbreviations	Company Cat. No.	Dilution for WES	Predicted Band (kDa) Based on Datasheet (WES)	Observed Band (kDa)/Peak on Chromatogram (WES)	Dilution for IF
Total histone H3	Total H3	#ab1791	1:100	17	26.9	1:1000
histone H3 mono methyl K4	H3K4me1	#ab239402	1:10	15	28.2	1:1000
histone H3 di methyl K4	H3K4me2	#ab32356	1:10	17	28	1:1000
histone H3 tri methyl K4	H3K4me3	#ab213224	1:10	15	28.3	1:500
histone H3 tri methyl K4 acetyl K9	H3K4me3K9ac	#ab272164	1:10	15	28.3	1:100
histone H3 phospho-S10	p-S10H3	#ab5176	1:10	15	25.0	1:300
GFP	GFP	#ab13970	*n.r*	*n.r*	*n.r*	1:2000

*n.r* means not relevant. WES, capillary Western immunoassay; IF, immunofluorescent staining.

## Data Availability

All data relevant to the study are included in the article or uploaded as online Appendix A.

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
