# Peer review of "Epigenetic Regulation and Molecular Mechanisms of Burn Injury-Induced Nociception in the Spinal Cord of Mice"

_ijms, 2024, doi:10.3390/ijms25158510_

Round 1
Reviewer 1 Report
Comments and Suggestions for Authors
The paper of Zoltan Meszar et all, entailed Epigenetic Regulation and Molecular Mechanisms of Burn Injury-Induced Nociception in the spinal cord of mice: explains burn injury mechanisms and uncovers converging and diverging pathways in pain models with different origins. The Authors identified 98 DEGs for burn injury and 86 DEGs for formalin-induced inflammatory pain.
The paper is both interesting and highly significant. Strengths of this paper include: a very interesting topic, a relevant and concise introduction; appropriate methodology, presented clearly; results displayed in easy-to-understand figures and tables; discussion that properly supports the results and references current literature; and conclusions drawn directly from the obtained results.
The reviewer has a few comments for the authors to consider:
1. Please add the section -future direction
2. After burn injury there is the persistence of the hypermetabolic state results in sustained loss of muscle mass and bone density. This hypermetabolic state, with a higher rate of protein degradation resulting in chronic amino acid loss that is sustained up to 1 year post-burn injury-whether this state may have influence on pain
3. Does the type of scar also may have influence on molecular mechanisms pain
Reviewer 2 Report
Comments and Suggestions for Authors
The authors present a study where RNA-seq analysis comparing burn injury and formalin-induced inflammatory pain identified distinct differences, with DEGs for burn injury and for formalin-induced pain, suggesting different central pain processing mechanisms. KEGG pathway analysis confirmed these differences, showing that burn injury activates Wnt signaling.
Methodologically it is a well-designed study, following the scientific method. It is adequately structured according to the standards of this journal. The images and tables provided are correct. The bibliographic references are current.
I have two questions about this well presented article. The first is to ask the authors to go deeper into the generation of the burns, as this model is crucial for the further development of the research. Did they perform histological analysis to determine the degree of burn depth in the paws of the mice?
Although this study indeed improves our understanding of burn injury mechanisms and reveals distinct pathways in different pain models, it is important to point out the possible clinical applicability of this study, so I encourage the authors to add a paragraph in the discussion on this topic.
